# Projecting the Potential Global Distribution of *Carpomya vesuviana* (Diptera: Tephritidae), Considering Climate Change and Irrigation Patterns

**Siwei Guo** [1] , **Xuezhen Ge** [2], **Ya Zou** [1], **Yuting Zhou** [1], **Tao Wang** [3] **and Shixiang Zong** [1,*]

[1]    Key Laboratory of Beijing for the Control of Forest Pests, Beijing Forestry University, Beijing 100083, China; siweiguooo@gmail.com (S.G.); zouyayaxx@gmail.com (Y.Z.); zhouyuting725@gmail.com (Y.Z.)

[2]    Department of Integrative Biology, University of Guelph, Guelph, ON N1G 2W1, Canada; xuezhen@uoguelph.ca

[3]    Mentougou Forestry Station, Beijing 102300, China; wtao315@126.com

*    Correspondence: zongshixiang@bjfu.edu.cn; Tel.: +86-13681389851; Fax: +10-62336073

**Abstract:** The ber fruit fly *Carpomya vesuviana* Costa (Diptera: Tephritidae) is the most destructive pests of *Ziziphus* spp. *Carpomya vesuviana* infestation causes great economic losses. We re-parameterized an existing CLIMEX model, and used the updated CliMond 30′ gridded resolution datasets within CLIMEX for the periods 1987–2016 and 2071–2100, representing historical and future climates, respectively, to predict the potential global distribution of the pest. Under the historical climate scenario, *C. vesuviana* had a wide climatically suitable distribution worldwide, from approximately 46° S to 50° N. Future climate change expanded the upper boundary of the potential distribution northward, and predicted that the pest would distribute approximately from 50° S to 60° N. Temperature was the primary determinant of the potential distribution of the pest among all driving variables. Irrigation was associated with a slight improvement in the climate favorability for the pest in some areas, including south-western North America, northern and southern Africa, and most of Oceania. The projections clarify the impacts of climate change on the potential global distribution of *C. vesuviana*, and are instructive for quarantine and management agencies for reducing economic damage caused by the fly and preventing expansion of *C. vesuviana* due to climate change.

**Keywords:** *Carpomya vesuviana*; CLIMEX; climate change; irrigation; potential distribution

## 1. Introduction

The ber fruit fly, *Carpomya vesuviana* Costa (*Carpomyia zizyphae* Agarwal & Kapoor, *Orellia bucchichi* Frauenfeld) (Diptera: Tephritidae), is regarded as the most serious pest of jujube (*Ziziphus* spp.) (Rhamanaceae) in southern Europe and central and southern Asia [1]. The species is native to India and has been found in approximately 20 countries, mainly in Asia, such as Iran, Oman, and Uzbekistan [2–6]. In addition to the increasing frequency of international trade, its small size, strong concealment ability, difficult quarantine enforcement, longevity, and reproductive ability are all putative explanations for its rapid spread [1].

As a monophagous pest that only infests *Ziziphus* spp., *C. vesuviana* has great economic implications for the jujube industry worldwide. It is the most devastating pest of the fruit of jujube trees, and has become a limiting factor for successful planting [2]. In Bushehr, Iran, it mainly harms *Z. spina-chresti* (L.) Desf., *Z. mauritiana* (Lamk.), *Z. nummularia* (Burm. f.) Wight & Arn., and *Z. lotus* (L.) Lam. [4]. In India, more than 100 pests damage jujube trees; as the most serious pest, *C. vesuviana* can jeopardize all kinds of jujube trees in the local area [7]. Adult *C. vesuviana* oviposit under the skin of jujube fruits. The larvae feed on the fruits, leading to early maturity and decay. Consequently, the infested fruits

become deformed and their growth is arrested [2], making the fruits unacceptable for the market. The *C. vesuviana* hazard rate can exceed 60%, reaching 73%–100% in Rajasthan, India [8]. The average yield loss of jujube fruit infested by *C. vesuviana* exceeds 40%, causing serious economic losses [9,10].

Global warming, characterized by an increase in ambient temperature, has been recognized as a major issue in the 21st century. According to the Fifth Assessment Report (AR5) published by the Intergovernmental Panel on Climate Change (IPCC), the average global temperature is projected to increase by 0.3–4.8 °C by the end of this century [11]. As poikilothermic animals, the change in temperature will have significant effects on all aspects of insect biology, including growth, reproduction, and geographical distribution [12,13]. To adapt to climate warming, insects tend to spread to higher latitudes (polarizations) or higher altitudes and colonize new territories [14]. Range expansions of harmful insects in response to climate change can cause significant losses to agriculture and forestry. In addition to global warming, other external environmental factors are important determinants of the distributions of insects. Irrigation, which influences humidity, will have various effects on the occurrence of insects for which the host plants rely on irrigation. Singh et al. [15] reported that heating of the soil by irrigation during the summer could kill the pupae of *C. vesuviana*. Yonow et al. [16] indicated that irrigation impacts the potential distribution of *Chilo partellus* (durra stem borer). Considering phenology and irrigation patterns, De Villiers et al. [17] predicted the potential distribution of *Bactrocera dorsalis*. These previous findings convey the importance of irrigation in predictive models.

Predicting changes in the potential distributions of species has become a major area of research, owing to the development of climate models to simulate future climate scenarios [18]. Two main research techniques are used to determine the potential geographical distribution of fruit flies: the more general niche model and the growth and development model. Four common niche models are used to predict species distributions: CLIMEX, DAVI-GIS, Maxent, and GARP. These models have already been applied to studies of fruit fly pests [19]. There have been several attempts to predict the habitat suitability of *C. vesuviana*. Lv et al. [20] used CLIMEX for a preliminary study of the potential distribution of the pest in China. He et al. [21] combined CLIMEX and GARP to establish a suitable distribution model of *C. vesuviana* in China, and subsequently improved the model [22]. Despite previous studies of the current distribution of *C. vesuviana* in China, accounting for irrigation, the potential distribution of the pest has not been predicted in future climate change scenarios. Therefore, it is necessary to identify areas of habitat suitability for *C. vesuviana* under climate change scenarios, considering irrigation patterns, to help decision-makers avoid the significant threat posed by the pests [23].

In this study, we used CLIMEX 4.0.0, combined with biological data and the known distribution, to identify the habitat favorability of *C. vesuviana* under both historical and future climate conditions. We also explored the effects of meteorological factors and irrigation on the occurrence and potential distribution of *C. vesuviana*. The projections will be instructive for the prevention and control of the pest.

## 2. Materials and Methods

### 2.1. Research Model and Software

#### 2.1.1. CLIMEX Model

CLIMEX 4.0.0 (Hearne Scientific Software, Melbourne, Australia) was used to predict the potential distribution of *C. vesuviana* across the world. CLIMEX provides a dynamic model that allows users to project the potential distribution and seasonal abundance of a species in relation to climate [24]. As the CLIMEX output, the annual eco-climatic index (EI) indicates the climatic suitability of a given location for the target species and is estimated by an annual growth index (GI) and a stress index (SI). For a more detailed introduction of CLIMEX, please refer to the study of Ge et al. [25].

### 2.1.2. ArcGIS Software

The Spatial Analyst Module of ArcMap 10.6, developed by the US Environment Systems Research Institute (ESRI) (RedLands, CA, USA), was used to transform the results obtained in the CLIMEX analysis by the function of inverse distance-weighted interpolation (IDW). Furthermore, the software was used to visualize the results and to calculate suitable areas for various categories of EI values for the species. Therefore, the potential global distribution of *C. vesuviana* could be intuitively and conveniently evaluated.

### 2.2. Data Collection

#### 2.2.1. Known Global Distribution of *C. vesuviana*

General data for the current global distribution of *C. vesuviana* were obtained from the literature, owing to the lack of relevant species records in various databases. Duplicate records and two records from the Global Biodiversity Information Facility (GBIF) without geographic coordinates were removed. Overall, *C. vesuviana* was mainly located in southern Europe and central and southern Asia.

#### 2.2.2. Climate Data

In CLIMEX, data for the following five meteorological parameters were required: monthly average minimum and maximum temperatures, monthly average precipitation, and average relative humidity at 09:00 h and 15:00 h. The CliMond 30′ gridded resolution datasets within CLIMEX for the periods 1987–2016 and 2071–2100 were used to represent historical and future climates, respectively. The historical climate data were obtained from version 4.01 of the gridded Climatic Research Unit (CRU) Time Series (TS) data for 1987–2016, with a spatial resolution of 0.5° latitude 0.5° longitude regular grid [26]. Future climate data were downloaded from the Coupled Model Inter-comparison Project Phase 5 (CMIP5). The Representative Concentration Pathways (RCP2.6, RCP4.5, RCP6.0, and RCP8.5) from CMIP5 describe four future greenhouse gas emissions conditions, considering economic, technological, demographic, policy, and institutional factors. RCP8.5 was used in this study, which is the most pessimistic estimate for the 21st century with respect to greenhouse gas emissions, consistent with no policy change to reduce emissions, a rapid increase in methane emissions, and heavy reliance on fossil fuels [27].

#### 2.2.3. Irrigaton Data

Irrigation has varying degrees of impact on species distributions and can influence soil moisture; accordingly, the effects of irrigation were considered in this study. Two irrigation scenarios were taking into account. First, an irrigation scenario of 1.5 mm day$^{-1}$ in the summer as top-up was applied globally to assess the threat posed by *C. vesuviana* in areas where jujube trees were sustained by irrigation (irrigation I). Second, a composite map was developed based on global irrigation areas reported by Siebert et al. [28]; in areas without irrigation, the EI for the natural rainfall scenario was mapped, and in areas under irrigation, the EI for the irrigation I scenario was mapped (irrigation II). The information on global irrigation areas was derived from the Food and Agriculture Organization of the United Nations (http://www.fao.org/home/en/index.html).

#### 2.2.4. Biological Data

To develop a better fit to the CLIMEX model of *C. vesuviana*, biological data for the pest were obtained from the literature, including information about temperature, humidity, and generations. These biological data were used to set the initial values of parameters (see Supporting Information, supplementary material 1 for details).

As for the life cycle, *C. vesuviana* develops in four stages through the life: egg, larva, pupa, and adult. Life cycle of the species varies with environmental factors [5]. According to Hu et al. [29],

the temperature has a significant effect on the developmental duration of the fruit fly. The fly can complete one generation successfully under a constant temperature of 20–32 °C, and the duration of each state shortens with the increase of temperature. Generally, the duration of the egg phase is 1–4 days with a viability of 70.21%–94.44%; the average durations of the larval and pupal phases are 6–22 and 8–320 days, respectively. And the period is long during December and short during March; the adult lasts 3 to 48 days in lab, which varies with month. 8–10 overlapping generations can be completed in one year [4]. 30 °C is the favorable temperature for pupal development and adult emergence, and pupation at 3 to 6 cm soil depth is ideal for adult emergence. Alternating rainfall of 20 to 40 mm and 62 to 85% relative humidity can also promote fly activity [5].

*2.3. Fitting Parameters*

CLIMEX parameters were fitted on the basis of biological data and geographic records of the species, considering environmental conditions that affect development and reproduction [24,30]. The parameter values published by He et al. [22] were used as a starting point to build the model, and were revised based on visual or experimental observations to obtain a better fit to the known distribution of *C. vesuviana*. In addition, an irrigation scenario of 1.5 mm day$^{-1}$ in the summer as top-up was applied to project the risk of *C. vesuviana* in areas where crops require irrigation. The most recent version of the Global Map of Irrigation Areas (GMIA) dataset was used to produce a composite map [28], including both irrigated and non-irrigated areas around the world, to show the overall projected suitability. The CLIMEX parameters that provide the best fit for the distribution of *C. vesuviana* at a global scale are summarized in Table 1.

**Table 1.** CLIMEX parameter values for *C. vesuviana.*

| Parameters | Descriptions | Values | | |
|---|---|---|---|---|
| | | **Lv et al. (2008)** | **He et al. (2011)** | **Current Model** |
| Moisture | | | | |
| SM0 | Lower soil moisture threshold | 0.1 | 0 | 0.028 |
| SM1 | Lower optimum soil moisture | 0.2 | 0.2 | 0.2 |
| SM2 | Upper optimum soil moisture | 0.85 | 0.4 | 0.4 |
| SM3 | Upper soil moisture threshold | 1.2 | 1.1 | 1.1 |
| Temperature | | | | |
| DV0 | Lower threshold | 7.7 | 13 | 13 |
| DV1 | Lower optimum temperature | 17 | 21 | 21 |
| DV2 | Upper optimum temperature | 30 | 36 | 35 |
| DV3 | Upper threshold | 39 | 40 | 41 |
| Cold stress | | | | |
| TTCS | Cold stress temperature threshold | 1.7 | −10 | −16 |
| THCS | Temperature threshold stress accumulation rate | −0.00008 | −0.00008 | −0.00008 |
| Heat stress | | | | |
| TTHS | Heat stress temperature threshold | 39 | 40 | 41 |
| THHS | Temperature threshold stress accumulation rate | 0.0005 | 0.005 | 0.005 |
| Dry stress | | | | |
| SMDS | Soil moisture dry stress threshold | 0.08 | 0 | 0.028 |
| HDS | Stress accumulation rate | −0.0007 | −0.00007 | −0.00007 |
| Wet stress | | | | |
| SMWS | Soil moisture wet stress threshold | 1.2 | 1.1 | 1.1 |
| HWS | Stress accumulation rate | 0.007 | 0.00003 | 0.00003 |
| Threshold heat sum | | | | |
| PDD | Number of degree-days above DV0 needed to complete one generation | 890 | 1100 | 800 |
| Irrigation scenario | | 3.5 mm day$^{-1}$ in winter | 0.3 mm day$^{-1}$ in April, 1.4 mm day$^{-1}$ in May and August, 2.8 mm day$^{-1}$ in June and July, 0.2 mm day$^{-1}$ in September | 1.5 mm day$^{-1}$ in summer as top-up irrigation |

### 2.3.1. Growth Indices (GI)

CLIMEX uses a weekly growth index scaled from 0 to 1 to describe conditions that favor population growth. The index is mainly a multiplicative combination of temperature and moisture indices [24,31].

### Temperature Index (TI)

The parameters were set according to experimental results and geographical records. DV0 and DV1 for population growth were left unchanged at 13 °C and 21 °C, respectively [22]. DV2 was set to 35 °C, to better represent the range of optimal temperatures for development reported previously [21,32,33]. He et al. [21] found that the optimum temperature for adult emergence is 25–35 °C, and the pupal duration is shortest at 33 °C. Ding et al. [33] reported that the most suitable temperatures for adult survival are 33–39 °C. Here, DV3 was increased to 41 °C, consistent with the results of Lakra et al. [2], who illustrated that an average weekly maximum temperature of 25–40 °C (the climate for local jujube fruit ripening in March and April) is optimal for *C. vesuviana*. Thus, the 40 °C figure in the original model of He et al. [22] was unlikely to be the upper threshold and was increased to 41 °C.

### Moisture Index (MI)

SM0 was set as low as possible (0.028) to indicate the permanent wilting point, which is normally about 2.8% of the soil moisture [34]. This allowed adequate growth to be simulated in the dry conditions of Xinjiang, China, where *Ziziphus jujube* grows well. SM1 and SM2 were set to 0.2 and 0.4 based on previous research showing that the pupae and adults of *C. vesuviana* prefer a soil relative humidity of 5%–25%; in addition, when the relative humidity of the air is 20%–40%, pupae and adults show a high survival rate and rapid development [33]. Similarly, SM3 was set to 1.1, which allowed for persistence in dry and wet conditions.

### 2.3.2. Stress Indices (SI)

The stress indices in CLIMEX are set to limit the potential growth of a population during adverse seasonal conditions. A value of 0 represents no stress in a given location, while a value of 100 reflects lethal conditions and precludes a species from persisting [24].

### Cold Stress (CS)

We decreased the cold stress temperature threshold (TTCS) to −16 °C, which was in accordance with the results of Ding et al. [35], who recorded that pupae have a mean supercooling point of −16.07 °C based on temperature gradient experiments. THCS was kept the same at −0.00008 week$^{-1}$, to ensure that it matched the distribution records in Xinjiang, China [36].

### Heat Stress (HS)

In this study, the heat stress temperature threshold (TTHS) parameter was increased to 41 °C to coincide with DV3. Additionally, THHS was maintained at 0.005 week$^{-1}$ to allow pest development in the known distribution.

### Dry Stress (DS)

Soil moisture dry stress threshold (SMDS) for DS was set to the same value as SM0 in our model to match the permanent wilting point of plants, at which plant growth ceases. HDS was unchanged at −0.00007 week$^{-1}$, which was adjusted based on *C. vesuviana* occurrences in some dry regions of India, Iran.

### Wet Stress (WS)

The WS indices were the same as those of the existing model of He et al. [22] Based on SM3, SMWS was set to 1.1, and HWS was set to 0.00003 week$^{-1}$ according to the humidity conditions tolerated by the species.

### 2.3.3. Effective Degree-Days (PDD)

In CLIMEX, the PDD degree-day parameter indicates the minimum thermal sum (degree-days above the minimum base temperature (DV0)) necessary for species to complete one generation [24,37]. Finally, the value of PDD was set to 800 degree-days to guarantee suitability in Bosnia, and to meet the actual situation of number of generations seen in India.

### 2.4. Classification of EI Values

The EI values are usually classified into four groups to show the climatic suitability of the species. Sutherst et al. [38] suggested that, in practice, EI values <10 indicate that a location is marginal for species survival, and values exceeding 20 support high population densities. Since the standard classification of EI values is species-dependent, it should be defined consistently with actual occurrences. However, it is difficult to classify the EI values of *C. vesuviana*, owing to the limited distribution records and lack of clear occurrence data. Therefore, we classified the EI values of marginal and moderate regions mainly based on the recommendations of Sutherst. According to previous reports, *C. vesuviana* is found in India and causes serious harm in Iran, especially in the regions of Busheh, Mashhad, Jask, Tehran, and Charbahar. Damage is also serious in Madurai, Srinagar, Bellary, and Bangalore of India [2]. To make the favorability of the potential distribution match the actual situation, the classification for EI in optimal and moderate regions was set to 23 after iterative debugging. Finally, the EI values were divided into four groups: unsuitable (EI = 0), marginal ($0 < \text{EI} \leq 10$), favorable ($10 < \text{EI} \leq 23$), and very favorable (EI > 23).

### 2.5. Parameter Verification

After running the final parameters in CLIMEX, the predicted results were reasonable. First, the modeled potential global distribution encompassed the known distributions very well. All occurrence records of *C. vesuviana* worldwide were within the suitable range in the model. Second, temperature and humidity in the predicted distribution were consistent with the climate of India, the native range of the species. In addition, the reasons for a lack of survival in some areas were consistent with the characteristics of the pest. Most of the unsuitable areas were concentrated in Russia and northern Asia, where the temperature is low. *C. vesuviana* prefers higher temperatures. Moreover, the numbers of generations of species in the distribution predicted by the model and the actual situation were consistent. In Turpan of Xinjiang, China, *C. vesuviana* exhibits two to three generations in a year; in northern India, it exhibits six–nine generations a year [2,21,39].

### 2.6. Analytical Results

Since *C. vesuviana* is an agroforestry pest, irrigation factors were considered. After comparing the results obtained using the two irrigation scenarios under historical climatic conditions, the composite (irrigation II) scenario was selected based on the similarity between the prediction and actual occurrences. By comparing areas with different EI values, the EI difference, and variation in EI among latitudes based on Ge et al. [25] under different irrigation scenarios, the impacts of irrigation were further analyzed. The driving meteorological variables were also explored. Analyses of how and where meteorological factors limit the potential distribution were performed. In addition, the difference in suitable areas under historical and future climatic conditions was evaluated, and the effects of future climate change on the species distribution with respect to the difference in the suitable range, habitat favorability, and climatic suitability at different latitudes were examined.

## 3. Results

### *3.1. Impacts of Irrigation on the Potential Distribution of C. vesuviana*

3.1.1. Potential Distribution for Two Types of Irrigation under Historical Climate Conditions

Under the historical climate, the potential global distribution in the absence of irrigation is shown in Figure 1a, and the distribution when assuming 1.5 mm day$^{-1}$ in the summer as top-up irrigation (irrigation I) is shown in Figure 1b. Based on areas under irrigation according to Siebert et al. [28] (Figure 1d), Figure 1c shows the composite (irrigation II) potential global distribution. There were significant differences in the potential distribution between the irrigation scenario and natural rainfall scenario, especially in northern Africa, western Asia, and Oceania. Many parts of northern Africa became suitable once irrigation was added, and most regions in Asia become more suitable after including irrigation. The difference in distribution between irrigation II and natural rainfall was slight, primarily involving areas in South America, southern Europe, southern Africa, south-east Asia, and northern Oceania. In particular, habitat favorability in these areas was stronger in the irrigation II scenario than in the natural rainfall scenario.

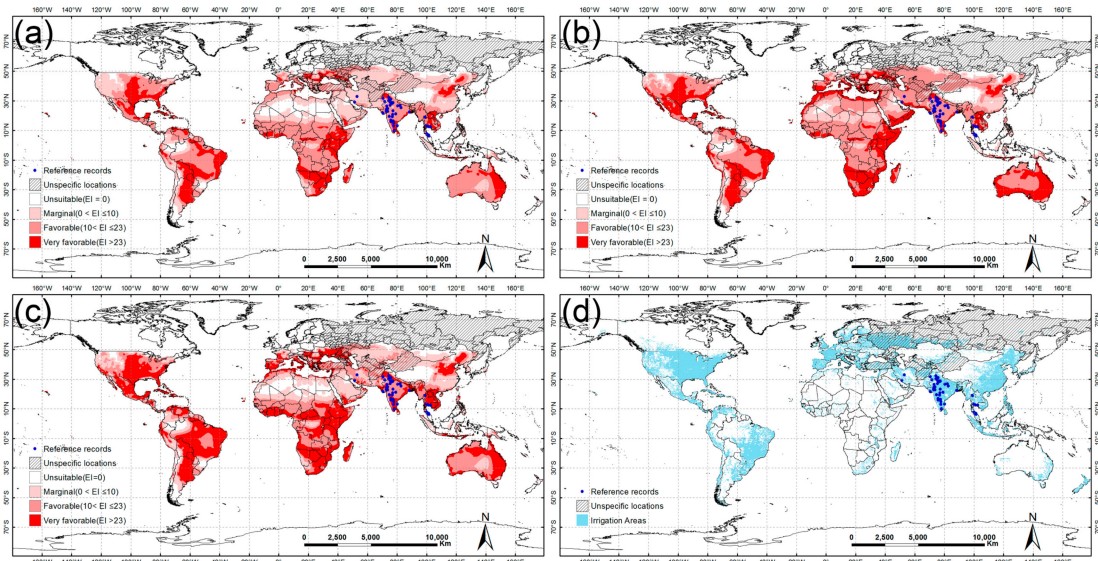

**Figure 1.** Potential global distribution of *Carpomya vesuviana* under the historical climate. (**a**) Projected global distribution of *C. vesuviana* assuming natural rainfall. (**b**) Projected global distribution of *C. vesuviana* with irrigation I. (**c**) Projected global distribution of *C. vesuviana* with irrigation II. (**d**) Global map of irrigation areas. The known current distribution is shown as reference records (blue dots) or as ambiguous locations (cross-hatched regions), which are indeterminate regions where specific record location(s) is/are unknown. White indicates unsuitable areas (EI (eco-climatic index) = 0); light pink indicates areas of marginal suitability (0 < EI ≤ 10); medium pink indicates areas of favorable suitability (10 < EI ≤ 23); red indicates areas of very favorable suitability (EI ≥ 23).

3.1.2. Comparison of the *C. vesuviana* Distribution for Two Types of Irrigation under Historical Climate Conditions

The potential distribution of *C. vesuviana* increased from 9.14% for very favorable areas without irrigation to 13.91% and 16.29% under the irrigation I and irrigation II scenarios, respectively (Figure 2). For the unsuitable and marginal habitats, the area in the absence of irrigation was larger than those in the two irrigation scenarios. The favorable habitat was greatest for irrigation I, with a narrower potential distribution for irrigation II than for a natural rainfall scenario.

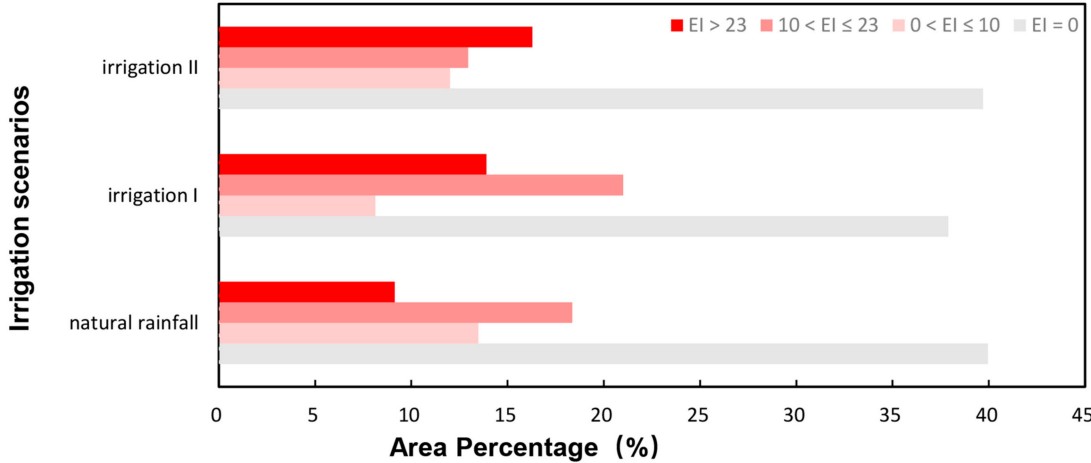

**Figure 2.** Changes in the potential distribution under historical climate conditions for two types of irrigation. Gray indicates unsuitable areas (EI = 0); light pink indicates areas of marginal suitability (0 < EI ≤ 10); medium pink indicates areas of favorable suitability (10 < EI ≤ 23); red indicates areas of very favorable suitability (EI ≥ 23).

Compared with the absence of irrigation, the suitability of the species increased when irrigation I was added (Figure 3a). Specifically, in south-western North America, the western coast of South America, northern and southern Africa, Spain, Turkey, and almost all of Oceania, the EI values increased; in south-west Asia, habitat favorability increased the most, and the increased rainfall was concentrated in northern Africa (Figure 3b). In addition, the difference between irrigation II and no irrigation (Figure 3c) was smaller than the difference between irrigation I and natural rainfall, but it still showed a wider potential distribution than that obtained for a natural rainfall scenario. Overall, the increase was concentrated in south-western North America, Peru, Algeria, Spain, Turkey, Iran, Pakistan, Afghanistan, Northwest China, Perth, and Melbourne, and the monthly average precipitation increased very little in northern Africa. We compared the difference in EI values between the irrigation I and irrigation II scenarios and the natural rainfall condition. As shown in Figure 3e, the difference was not very large. Overall, the area where the pest can adapt at all latitudes increased, and the EI increased from about 0° N to 50° N and 45° S to 14° S. The main differences were from 30° S to 25° N; favorability under irrigation II in these areas increased less than favorability under the irrigation I scenario, with almost no change at all.

The different irrigation scenarios resulted in a slight improvement in the potential distribution compared with that of the natural rainfall scenario, except for marginal habitats, which were larger in the absence of irrigation and only differed by 2.38% between the irrigation I and irrigation II scenarios. Although the suitable conditions in the irrigation I scenario were better, the results did not match the actual situation in arid Africa, considering the global irrigation range (Figure 1d); we ultimately added the irrigation II pattern to predict the effects of climate change on the potential distribution of the pest.



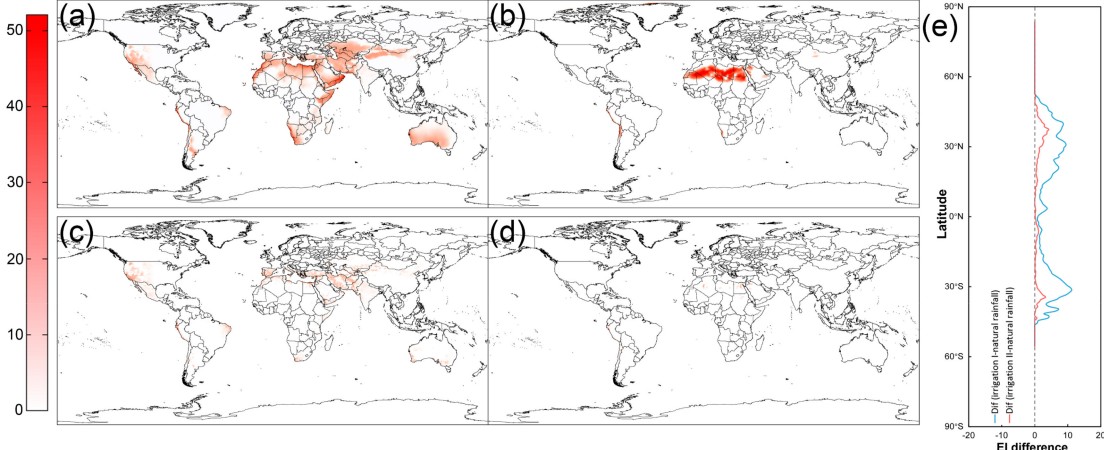

**Figure 3.** Impact of different irrigation scenarios on climatic suitability and average monthly precipitation. (**a**) Difference between irrigation I and natural rainfall conditions on climatic suitability. (**b**) Difference between irrigation I and natural rainfall conditions on average monthly precipitation. (**c**) Difference between irrigation II and natural rainfall conditions on climatic suitability. (**d**) Difference between irrigation II and natural rainfall conditions on the average monthly precipitation. Red indicates an increase; depth of color indicates the degree of change in EI values and average monthly precipitation. (**e**) Changes in EI difference with increasing latitude; the blue line represents the difference between irrigation I and natural rainfall; the red line represents the difference between irrigation II and natural rainfall.

*3.2. Potential Global Distribution of C. vesuviana under Different Climate Conditions*

3.2.1. Potential Global Distribution of *C. vesuviana* under Historical Climate Conditions

The historical potential global distribution of *C. vesuviana* is shown in Figure 4a. *C. vesuviana* had a wide climatically suitable distribution from about 46° S to 50° N, with a total area of $75.91 \times 10^6$ km$^2$, accounting for 50.95% of the world's land mass (excluding Antarctica).

The area of marginal habitats covered $22.10 \times 10^6$ km$^2$, accounting for 29.11% of the total potential distribution, including large regions (Figure 4a) in Asia (Kazakhstan, southern Mongolia, north-western and south-eastern China, central Iran, Saudi Arabia, and Oman), and scattered areas of the north-western United States, Colombia, and northern Africa. The favorable habitat was $23.84 \times 10^6$ km$^2$, which represented 31.40% of the total potential distribution. This habitat was mainly located in Asia (central India, western and south-western Iran, Turkey), Africa (Nigeria, Congo), and Oceania (western Australia). There were also sporadic locations in China, Kazakhstan, the United States, and Brazil. The area of very favorable habitat amounted to $29.98 \times 10^6$ km$^2$, which was 39.50% of the total potential distribution; it was distributed mainly in North America (the United States, Mexico), South America (Venezuela, Brazil, Argentina), Africa (large parts of southern Africa, Madagascar), Asia (Thailand, Laos, north-eastern China), and Oceania (Queensland, New South Wales).

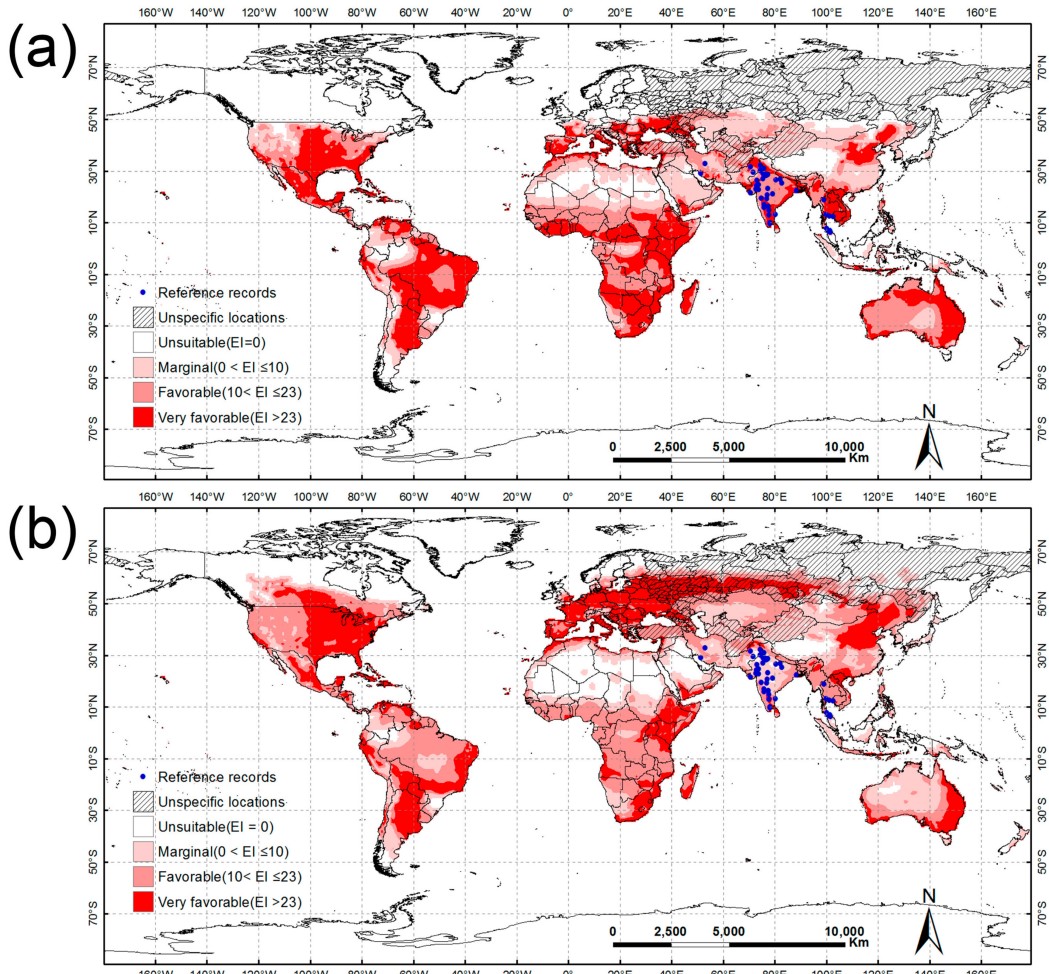

**Figure 4.** Potential global distribution of *C. vesuviana* with the irrigation II scenario. (**a**) Potential global distribution of *C. vesuviana* under historical climate conditions. (**b**) Potential global distribution of *C. vesuviana* under future climate conditions. Known current distribution is shown as reference records (blue dots) or as ambiguous locations (cross-hatched regions), which are indeterminate regions in which specific record location(s) is/are unknown. White indicates unsuitable areas (EI = 0); light pink indicates areas of marginal suitability (0 < EI ≤ 10); medium pink indicates areas of favorable suitability (10 < EI ≤ 23); red indicates areas of very favorable suitability (EI ≥ 23).

### 3.2.2. Potential Global Distribution of *C. vesuviana* under Future Climate Conditions

Under future climate conditions, the predicted global potential distribution of *C. vesuviana* is shown in Figure 4b. In general, climate change expanded the upper boundary of the potential distribution northward, from approximately 50° N to 60° N. The total suitable potential distribution was predicted to increase to $91.77 \times 10^6$ km$^2$, which is about 61.59% of the world's land mass (excluding Antarctica).

We compared the overall distributions in the future change scenario and the historical climate conditions (Figure 4b). The potential areas of distribution were projected to increase significantly in Europe and Asia, changed little in North America and South America, and decreased slightly in Africa. In particular, the total area of unsuitable habitats decreased, despite increases in some regions of South America, northern Africa, and Oceania. Marginal habitats were mainly located in Oceania, which was originally favorable. Favorable habitats increased substantially in southern Africa, North America, South America, and central Asia; most of these areas were transformed from very favorable areas. In addition, the areas in Oceania that had been favorable in the past typically became marginal habitats. The very favorable habitat was concentrated in Europe and Central Asia, and areas that were originally

very favorable exhibited varying degrees of change. The climate in most regions may become less suitable for pest survival in the future.

### 3.2.3. Comparison of Distributions under Current and Future Climate Conditions

Under future climate conditions, the total suitable potential distribution was predicted to increase by $15.85 \times 10^6$ km$^2$. The areas of marginal and favorable habitats were predicted to increase by $3.53 \times 10^6$ km$^2$ and $13.91 \times 10^6$ km$^2$, accounting for approximately 22.24% and 87.77% of the total increase, respectively. The predicted area of very favorable habitat decreased slightly by about $1.58 \times 10^6$ km$^2$ (Figure 5). The increase in the total suitable distribution mainly reflected an increase in favorable habitats.

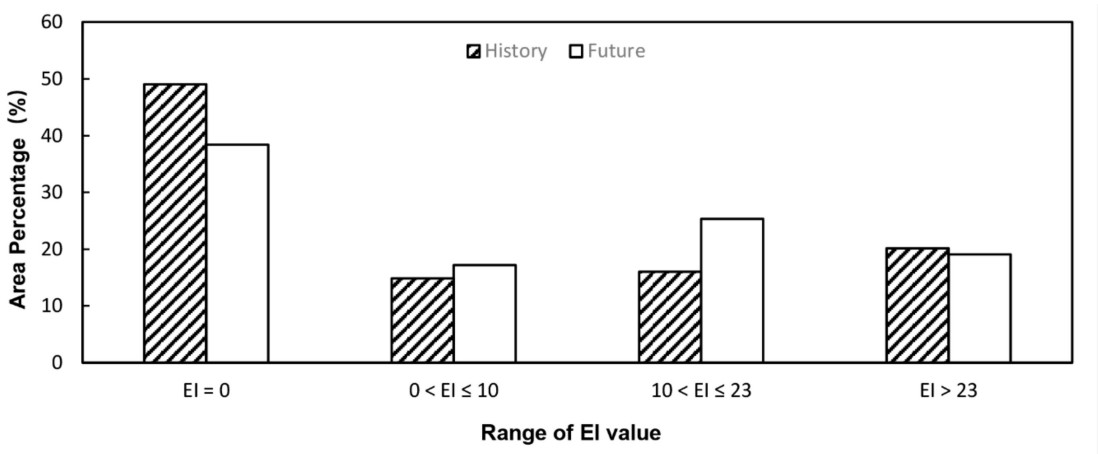

**Figure 5.** Area proportions of different ranges of EI values for *C. vesuviana* under historical and future climate conditions.

We compared the difference in EI values under historical and future climate scenarios, and mapped the change in habitat favorability (Figure 6). The impacts of climate change on suitability (EI difference) for *C. vesuviana* would differ among latitudes (Figure 6b). The increase in EI was concentrated at 30° S to 50° S and 30° N to 60° N, and the EI value for the area from 30° S to 30° N decreased, indicating a reduced fitness. Under future climate conditions, areas of favorability were predicted to decrease to different extents (Figure 6a), like in northern South America, southern Africa, southern Asia, and Oceania. The opposite results were obtained for most parts of central North America, Europe, and northern Asia.

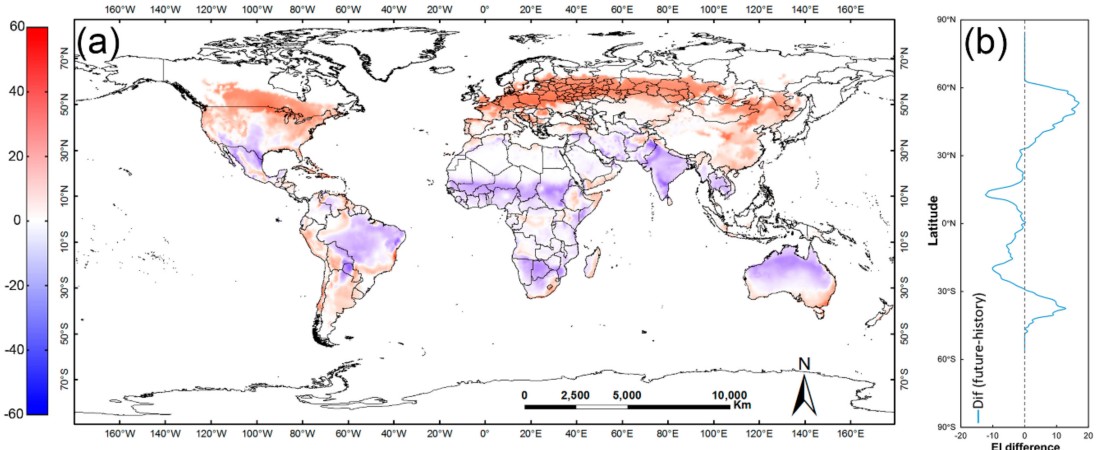

**Figure 6.** Impact of climate change on climatic suitability. (**a**) Differences in EI values between historical and future climatic conditions. Red indicates an increase, blue indicates a decrease, and depth of color indicates the degree of change in EI values. (**b**) Changes in EI difference with increasing latitude.

*3.3. Driving Variables Limiting the Potential Distribution*

Based on the EI formula, which integrates the annual growth index (GI) with annual stresses that limit survival during the unfavorable season and any limiting factors, the survival of *C. vesuviana* will be limited for a GI value of 0 or a SI greater than 100. Thus, in our model, the EI values for *C. vesuviana* are mainly affected by the six variables, TI, MI, CS, HS, DS, and WS, that limit the potential distribution.

The limits of the potential global distribution based on the variables driving *C. vesuviana* populations are shown in Figure 7. In both historical and future conditions, TI was the main factor limiting the distribution in north-eastern Canada and Greenland. It was also important in scattered areas in the northern border region of Russia and south-west China under the historical climate. In parts of southern Greenland, northern Africa, Indonesia, Papua New Guinea, south-western South America, and eastern Canada, EI was zero in the absence of MI. In northern Africa, the potential distribution was mainly limited by HS and DS (Figures 7 and 8); CS was the main limiting factor for parts of Greenland. WS limited population growth in Indonesia and north-western South America. These limiting effects were consistent with the results of EI (Figure 4).

In addition, climate change affected some variables that influenced the potential distribution of species. In the future, TI values in northern Canada and central Asia would increase, consistent with an increase in suitable areas in these region (Figure 9). The combined effect of HS and TI in northern Africa (HS increased and TI decreased) under future climatic conditions made the region no longer suitable. In CLIMEX, the PDD degree-day parameter indicates the minimum thermal sum (degree-days above the minimum base temperature (DV0)) necessary for species to complete generations [24,37]. Finally, the value of PDD was set to 800 degree-days to fit the suitability in Bosnia; in addition, the 800 degree-days allowed for modeling of the actual situation of number of generations in India.

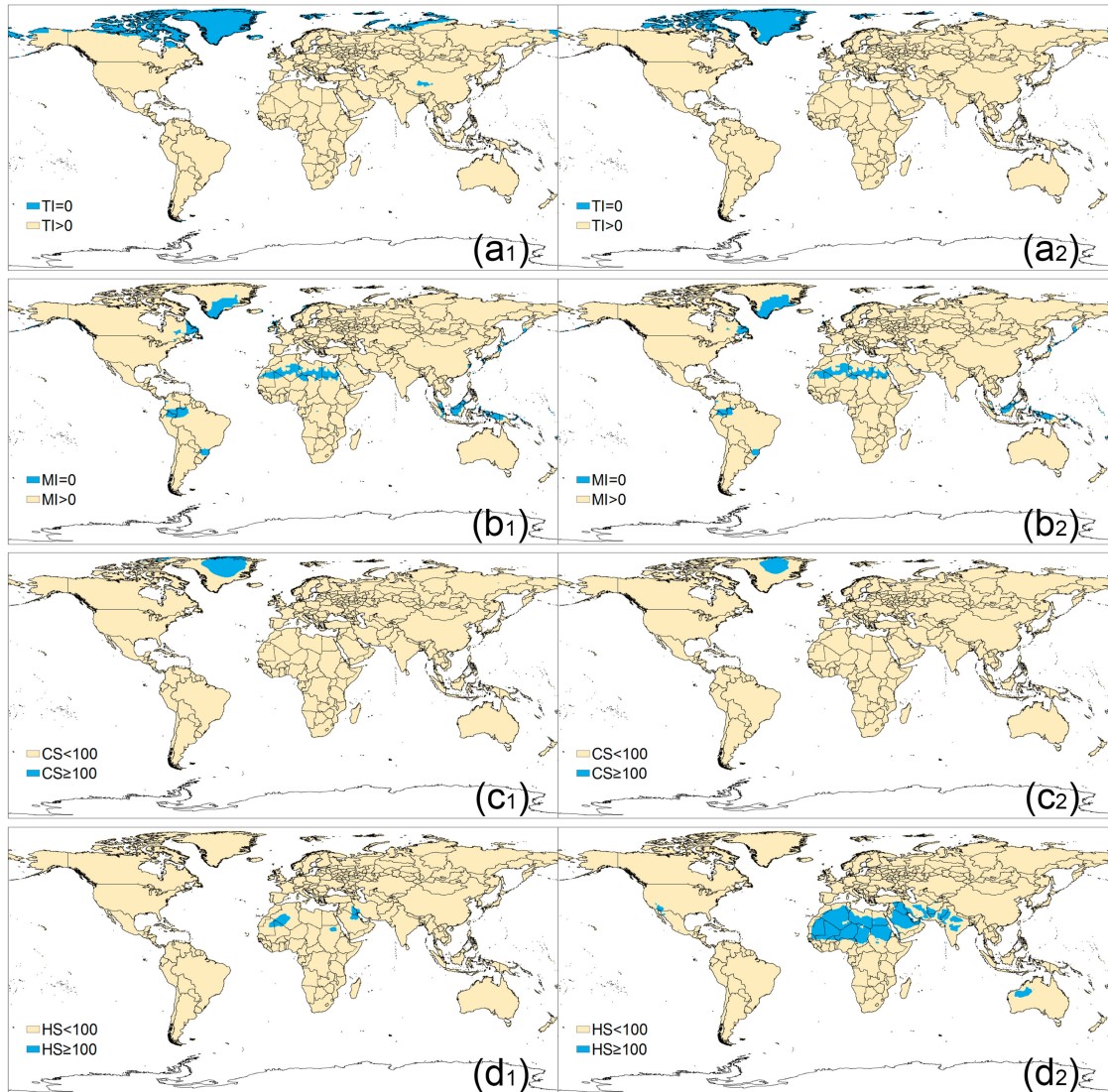

**Figure 7.** Limiting distribution maps for four different conditions. (**a**) Regions in which the temperature index (TI) will be unsuitable for survival are shown in blue. (**b**) Regions in which the moisture index (MI) will be unsuitable for survival are shown in blue. (**c**) Regions in which cold stress (CS) will be unsuitable for survival are shown in blue. (**d**) Regions in which heat stress (HS) will be unsuitable for survival are shown in blue. Subscript (1) indicates historical climatic conditions and subscript (2) indicates future climatic conditions.

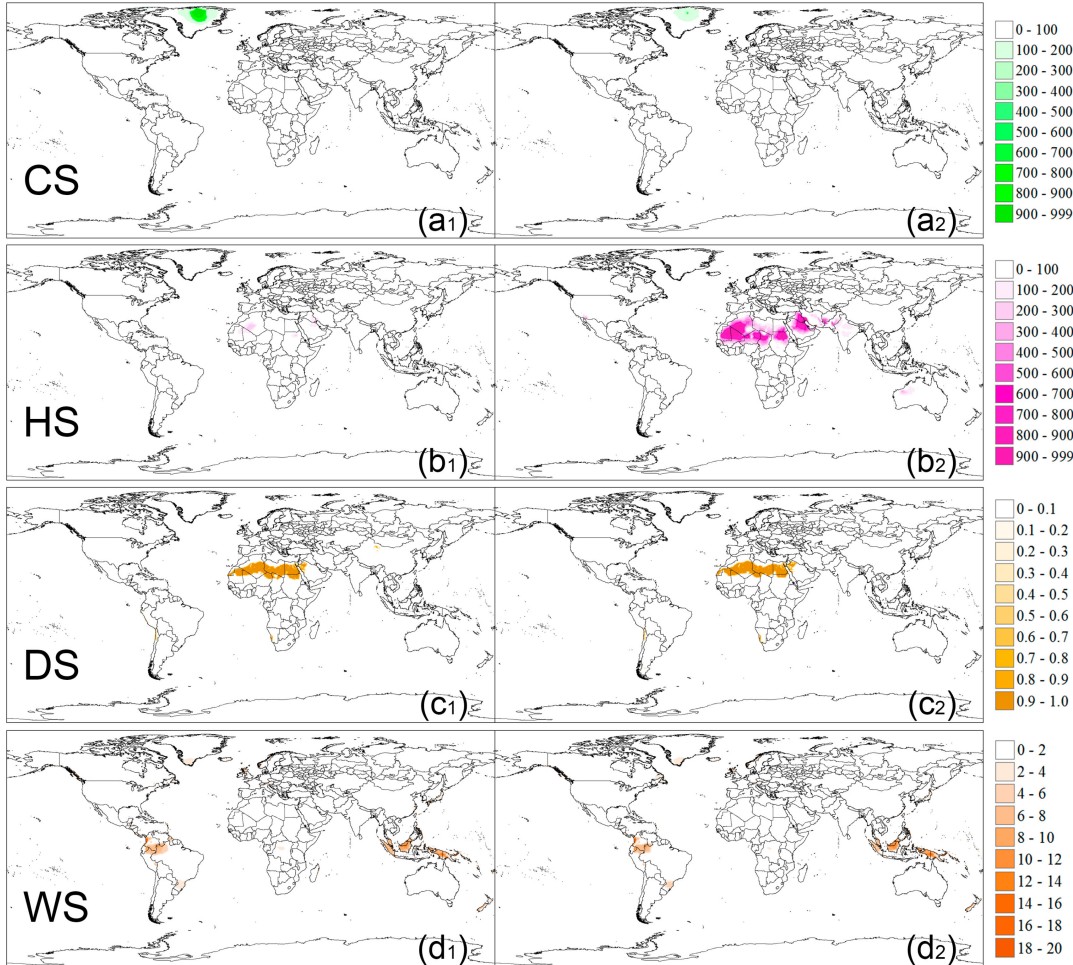

**Figure 8.** Global map of monthly stress indices for *C. vesuviana*. (**a**) Map of monthly cold stress (CS), (**b**) map of monthly heat stress (HS), (**c**) map of monthly dry stress (DS), and (**d**) map of monthly wet stress (WS). Subscript (1) indicates historical climatic conditions and subscript (2) indicates future climatic conditions. Depth of color indicates the weight of each value: the darker the color, the larger the value.

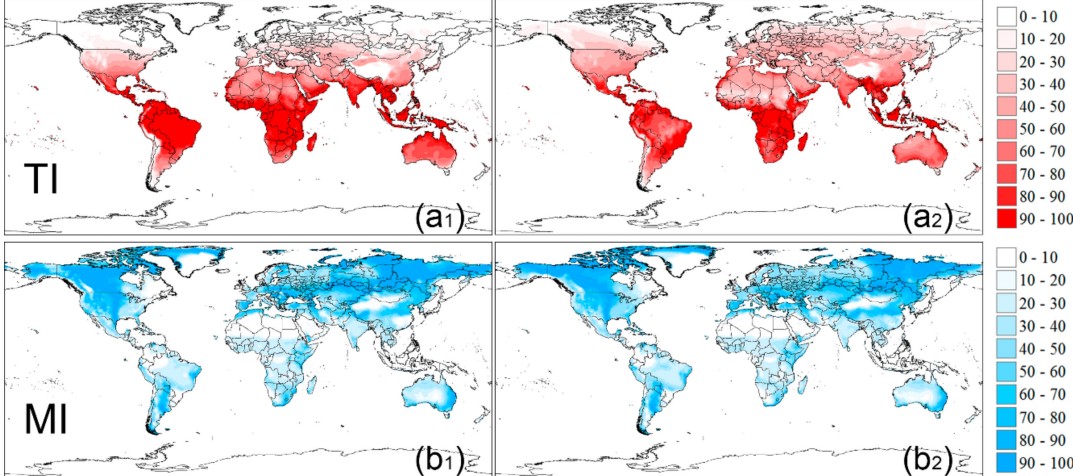

**Figure 9.** Global map of monthly growth indices for *C. vesuviana*. (**a**) Map of the monthly temperature index (TI), and (**b**) map of the monthly moisture index (MI). Subscript (1) indicates historical climatic conditions and subscript (2) indicates future climatic conditions. Depth of color indicates the weight of each value: the darker the color, the larger the value.

## 4. Discussion

Here, we compare the present results with those of previous studies. Compared with the results of our model, the model of Lv et al. [20] showed that much of south-western Russia was estimated to be suitable (Figure S1a), where *C. vesuviana* was known to occur, and that it will have suitable areas from 55° S to 70° N in the future. We projected essentially the same range of suitable areas as He et al. [22], from 50° S to 60° N, except for most of southern Canada and parts of Russia, which we projected to be suitable. He et al. [22] predicted that more areas in northern Africa will support species survival. We think that differences in parameter settings, data sources (known distribution records, biological data, and meteorological data), and analytical methods can explain the differences among the three models. Specifically, we used recent records and research focused on the biological characteristics of *C. vesuviana* to guide parameter settings. Differences in EI values among studies may contribute to differences between the potential distributions. In addition, the meteorological data used by the two previous studies were the same (historical climate data were obtained from 748 meteorological stations in China from 1971 to 2000, provided by the National Meteorological Information Center). We used the CRU Time Series data for 1987–2016, and downloaded future climate data (2071–2100) from the CMIP5. With regard to analytical methods, all three studies divided EI values into four levels. Lv et al. [20] categorized values as unsuitable (EI = 0), marginally suitable (0 < EI < 10), suitable (10 ≤ EI < 20), and very suitable (EI ≥ 20). He et al. [22] divided EI values as follows: unsuitable (EI = 0), marginally suitable (0 < EI ≤ 10), moderately suitable (0 < EI < 30), and highly suitable (EI ≥ 30); He et al. [22] defined the overlap in the adaptive prediction results of CLIMEX and GARP as the main climate-friendly areas, non-overlapping parts as secondary suitable areas, and the other areas as unsuitable. Therefore, differences in the method for categorizing EI values can explain differences in suitability and prediction results. Moreover, since current research by He and Lv is focused on China, their parameters may be unsuitable for predicting the global potential distribution of *C. vesuviana*.

Our results showed that climate change will have different effects on *C. vesuviana* in different regions of the world, with increases in insect suitability in some regions and decreases in others. Based on the observed impacts of six meteorological factors and irrigation, as well as biological characteristics, we speculate on the factors that explain changes in the species distribution.

Climate change will affect meteorological factors, and then act on the potential distribution of species. According to the results shown in Figure 6, Figures S2 and S3, over time, the climate is likely to become more favorable in North America, due to the combined effects of temperature and relative humidity. Climate change will increase both TI and MI values and decrease WS in parts of eastern Canada, thereby extending suitability to northern North America. The species originated in India and, thanks to its high temperature-resistant biological properties, *C. vesuviana* may be more likely to survive in high-temperature environments. Accordingly, in northern South America, central Africa, south-western Asia, and northern Oceania, temperature decreases can explain the decreases in EI values. In southern South America, central Asia, and most of Europe, due to the increase in TI, suitable habitats will increase. Even though *C. vesuviana* originated in arid regions, the species may prefer a humid environment. Thus, the climate in some parts of northern Asia will become less favorable over time, which may essentially be due to the decrease in humidity, despite a suitable temperature. The increase in CS in our model limits the northward expansion of *C. vesuviana* to 60° N, and northern Russia, northern North America, and Greenland are unsuitable according to the model. In addition to temperature, HS increases due to climate change, which will limit the survival of the species in most parts of northern Africa. This factor will also affect the south-western part of Asia.

Irrigation affects the distribution of species, equivalent to changing the amount of precipitation in irrigated areas. We evaluated the effects of irrigation on species occurrence in this study by adding irrigation data to the precipitation data. We used monthly average precipitation data for each meteorological site, derived from CLIMEX. Under the irrigation II scenario, we changed the precipitation data in areas where the precipitation was below the irrigation level of 1.5 mm day$^{-1}$ (i.e., 45 mm month$^{-1}$), to the same as the condition of the model as the precipitation data under the irrigated

region; the precipitation data in the corresponding natural rainfall model were used as precipitation in non-irrigated areas. As shown in Figure S4, in some areas, such as central North America and central and south-eastern Asia, suitability will increase with increasing precipitation. In southern North America and southern Africa, the favorability in these areas will decrease due to reduced precipitation. However, in south-western Asia, precipitation increased while predicted suitability decreased, which can probably be attributed to increasing heat stress. Precipitation did not change substantially in Oceania, and EI decreased, possibly as a consequence of the decrease in temperature mentioned above. Therefore, it is likely that temperature and humidity have the greatest impact among factors that limit *C. vesuviana* suitability. In addition, the species may prefer high temperature and humidity environments, such as southern North America, eastern Oceania, and southern Asia; cold, low humidity, and CS will restrict survival to a greater degree, like in Greenland and the northern part of Asia.

The irrigation scenarios we applied in this study have some limitations. First, we were only able to identify where irrigation has been applied in the world, but were not able to obtain specific data for monthly irrigation. The irrigation data we used were only based on the actual and predicted distributions under irrigation conditions in our model and its predecessors. Second, in the irrigation II scenario, the statistics for global irrigation areas may have been updated. In addition, irrigation under future and historical conditions may not be completely consistent. These issues may impact the study outcomes and predicted distributions.

We identified areas that might become vulnerable to invasion by *C. vesuviana* as a result of climate change and irrigation patterns. These results are useful for making recommendations to address pest invasion, and for preparing strategies towards minimizing economic impacts in high-risk areas and for the long-term quarantine of this pest as a result of climate change. However, the prediction of suitable areas was based solely on climatic factors and does not incorporate biotic interactions (host and predator availability), land use, soil type, and human activities, which will also affect the dispersal, survival, and establishment of the species [40,41]. Therefore, in future studies, it will be necessary to include additional non-climatic factors in CLIMEX models, and to combine multiple models to reduce uncertainty and improve prediction accuracy for a more comprehensive view.

## 5. Conclusions

In summary, the potential distribution of *C. vesuviana* will be wider with climate change. The suitability will increase in middle latitude regions, while it will decrease in low latitude regions. Temperature was the primary determinant of the potential distribution of the pest, and irrigation had slight effects on climate favorability. The projections for potential distributions are instructive for quarantine and management agencies to reduce economic damage caused by *C. vesuviana* and prevent expansion of the pest due to climate change.

**Supplementary Materials:** The following are available online at http://www.mdpi.com/1999-4907/10/4/355/s1, Figure S1: Projected potential distribution using existing CLIMEX parameters for *C. vesuviana*; Figure S2: Impacts of climate change on monthly growth indices for *C. vesuviana*; Figure S3: Impacts of climate change on the monthly stress indices for *C. vesuviana*; Figure S4: Impacts of climate change on the precipitation and climatic suitability; Supplementary Material 1: Biological data related to temperature, moisture, and generations of *C. vesuviana*.

**Author Contributions:** Conceptualization, X.G. and S.Z.; Data curation, S.G., Y.Z. (Ya Zou), Y.Z. (Yuting Zhou) and T.W.; Formal analysis, S.G., Y.Z. (Ya Zou) and Y.Z. (Yuting Zhou); Funding acquisition, S.Z.; Investigation, S.G.; Methodology, S.G., X.G. and S.Z.; Project administration, X.G. and S.Z.; Resources, T.W.; Software, S.G.; Supervision, X.G. and S.Z.; Validation, Y.Z. (Ya Zou) and Y.Z. (Yuting Zhou); Writing—original draft, S.G. and X.G.; Writing—review & editing, X.G. and S.Z.

**Funding:** This study was supported by "the Fundamental Research Funds for the Central Universities" (No. 2016ZCQ07).

**Conflicts of Interest:** The authors declare no conflict of interest.

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
