# Peer review of "Projecting the Potential Global Distribution of Carpomya vesuviana (Diptera: Tephritidae), Considering Climate Change and Irrigation Patterns"

_forests, doi:10.3390/f10040355_

Round 1
Reviewer 1 Report
It is an interesting study suggesting the predicted global distribution of the Carpomya vesuviana pest. It is well written and described.
I have read the MS a couple of times and I do not find any major problem
with it. I have seen few typos or formatting issues but no spelling
mistakes so did not report it. As I assume the editor would suggest the
formatting changes. This paper is a good read.
If anything I would suggest adding few lines on the biology of the pest
understudy such as life cycle and life history study like what
temperature it prefers for the growth and reproduction as the whole
paper talks about the climatic conditions.
Author Response
Dear Reviewer:
We really appreciate for your valuable suggestions on our manuscript, and we have studied your comments carefully. All revised portions are marked using the "Track Changes" function in the revised manuscript. Responses to reviewer’ comments are listed as blow. And once again, thanks for all your dedicated work.
Best regards!
Sincerely yours,
Shixiang Zong

Reviewer 2 Report
changes are tracked in PDF file. please incorporate.

Author Response
Dear Reviewer:
We really appreciate for your kind suggestions on our manuscript, and we have studied the comments carefully. All revised portions are marked using the "Track Changes" function in the revised manuscript. Responses to reviewer’ comments are listed in the cover letter. And once again, thanks for all your dedicated work.
Best regards!
Sincerely yours,
Shixiang Zong

Reviewer 3 Report
Review of Guo et al
Projecting the potential global distribution of Carpomya vesuviana (Diptera: Tephritidae), considering climate and irrigation patterns
This is a generally well-written paper that models the potential distribution of Carpoyma vesuviana using CLIMEX under climate change and irrigation scenarios. Other researchers (some of which the authors did not cite) have done the exact same thing with other insects, so from that standpoint this work is not original.
The CLIMEX methods are also standard (already described dozens of times in the entomology literature alone) and do not require originality. Table 1 is the standard CLIMEX template and its parameters are well known. I wonder if the authors can just describe how they obtained the values, as they have, and not define every parameter in the text. Rather, they could just refer to other papers for definitions.
This is the first time it has been done for this fly and the results are significant and compelling enough for publication. There is very small production of jujubes in the U.S. in California and Texas, so far free of fly infestation.
Some select (not all) editorial changes needed:
L14: Spell out Carpomya
L14: …causes economic losses. Delete “and ecological losses wordwide.”
What ecological losses? And “worldwide” is an exaggeration.
L19: upper boundary
L24: delete “etc.” don’t use “etc.” in a scientific paper
L25-26: agencies for reducing economic damage caused by the fly and preventing expansion of…
Delete “ecological damage”
L30: jujube (Ziziphus spp.) (Rhamanaceae)
L32: Asian, including Iran, Oman.. delete “etc”
L34: what is strong survival – just use “longevity”
L36: infests, not infects
L37: pest of the fruit of jujube trees
L44-45: This is not a complete sentence and needs to be re-written
L53: can cause significant losses to agriculture and
L54: delete without control
L67: attempts to predict
L339: the upper boundary of the
L389: generations [24, 25] (delete the period)
L398: use a word other than “guarantee”
L399: in Bosnia. In addition, the 800 degree-days allowed for the actual …
Discussion
L418: Start the paragraph with a topic sentence. Here, it could be a comparison of present results with those of previous studies.
L418: results of our model, the model of Lu et al. [20] showed that much of southwestern Russia was suitable..
L435: they defined
L448: The species originated in India and has high ability to tolerate high temperatures and may be
Delete “has developed into a disaster” which sounds silly
L441-477: This paragraph is way too long, rambles incoherently, and needs to be broken up into smaller paragraphs, each with its own topic sentence. Irrigation needs to be its own paragraph, for instance. Paragraph needs a lot of work to make it the best it can be.
L485: The authors did not show that “C. vesuviana poses a clear danger to jujube production”, so DELETE this
Just state that “We identified areas that might become vulnerable to invasion by C. vesuviana as a result of climate change and irrigation patterns.”
Did not thoroughly look at all references (there might be bunch of errors), but:
20. Lv, W. G.
Is this Lu et al?
28: italicize Schizaphis gramimum
Author Response
Dear Reviewer:
We really appreciate for your kind advice on our manuscript, and we have studied the comments carefully. All revised portions are marked using the "Track Changes" function in the revised manuscript. Responses to reviewer’ comments are listed in the cover letter. And once again, thanks for all your dedicated work.
Best regards!
Sincerely yours,
Shixiang Zong
